

# Enhancing breast cancer diagnosis using deep learning and gradient multi-verse optimizer: a robust biomedical data analysis approach

Yassine EL kati, Shu-Lin Wang, Mundher Mohammed Taresh and Talal Ahmed Ali Ali

College of Computer Science and Electronic Engineering, Hunan University, Changsha, Hunan, China

## ABSTRACT

Breast cancer (BC) is one of the most common causes of mortality among women. However, early detection of BC can effectively improve the treatment outcomes. Computer-aided diagnosis (CAD) systems can be utilized clinical specialists for accurate diagnosis of BC in its early stages. Due to their superior classification performance, deep learning (DL) methods have been extensively used in CAD systems. The classification accuracy of a DL model mainly depends on the parameters, such as weights and biases, of the deep neural network (DNN), which are optimized during the training phase. The training of DL models has been carried out by gradient-based techniques, *e.g.*, stochastic gradient descent with momentum (SGDM) and adaptive momentum estimation (ADAM), and metaheuristic techniques, *e.g.*, genetic algorithms (GA) and particle swarm optimization (PSO). However, these techniques suffer from frequent stagnation in local optima due to the huge search space, which can lead to sub-optimal DL performance. This article proposes a hybrid optimization algorithm, based on incorporating a simple gradient search mechanism into a metaheuristic technique, multi-verse optimizer (MVO), to facilitate the search for global optimal solution in the high-dimensional search space of DL models. A DL model for BC diagnosis is developed based on a three-hidden-layer DNN whose parameters are optimized using the proposed hybrid optimizer. Experimental analysis is carried out on the Wisconsin breast cancer dataset (WBCD) and the Wisconsin Diagnosis Breast Cancer (WDBC) dataset, each is divided into 70% for training and 30% for testing. For comparison reasons, similar DL models trained using various optimizers, including gradient-based, metaheuristic, and recently-proposed hybrid optimization algorithms, are also analyzed. The results demonstrate the superior performance of our optimizer in terms of attaining the most accurate DL model in the fastest convergence rate. The proposed model achieves outstanding metrics, including accuracy at 93.5%, precision at 88.06%, specificity at 93.06%, sensitivity at 95.64%, F1 score at 91.67%, and Matthew's correlation coefficient (MCC) at 87.14% on WBCD, and accuracy at 96.73%, precision at 93.38%, specificity at 95.83%, sensitivity at 98.25%, F1 score at 95.75%, and MCC at 93.18% on WDBC, in just six epochs. This research significantly contributes to advancing CAD systems for BC, emphasizing the potential benefits of the proposed optimizer in medical classification domains.

Corresponding authors
Yassine EL kati,
elkati.yassine@proton.me
Shu-Lin Wang,
smartforesting@163.com

## INTRODUCTION

Breast cancer (BC) poses a significant threat with high incidence and mortality rates among women. The complex nature and unclear causes of BC make its cure is challenging (*Tinoco et al., 2013*; *Sun et al., 2017*). Fortunately, early detection of BC can significantly enhance treatment outcomes and increase survival rates. In the classical approach, the diagnosis process of BC highly depends on the accessibility of a rigorous examination (*e.g.*, the triple assessment test (*Nigam & Nigam, 2013*)) as well as the experience and visual inspections of the in-charge physician. However, such a diagnosis approach is prone to error and time consuming.

Recently, computer-aided diagnosis (CAD) approach based on artificial intelligence has been introduced as a potential alternative to the classical approach and utilized to detect BC and to classify it as benign or malignant. Many CAD systems have been constructed based on traditional machine learning (ML) techniques, such as support vector machine (SVM), K-nearest neighbors (KNN), and decision trees (DT) (*Elkorany et al., 2022*; *Uddin et al., 2023*). The manual feature extraction in such systems is time consuming and may degrade the classification performance. With the success of deep learning (DL) over traditional ML, many DL methods have been successfully employed for automatic BC diagnosis, *e.g.*, convolutional neural network (CNN)-, recursive neural network (RNN)-, and data belief network (DBN)-based approaches (*Kaur, 2023*; *Neffati & Machhout, 2023*; *Archana, 2024*). Unlike traditional ML, DL supports automatic feature extraction and can learn new features itself thanks to the hierarchical design of deep neural networks (DNNs). This makes DL models more powerful and require fewer human interventions.

This article deals with the problem of training a DL model, *i.e.*, obtaining the model's parameters such as weights and biases, for optimal BC diagnosis. Although DL methods are promising, the training process of a DL model is a challenge due to the highly nonlinear model, the large number of parameters, and the need for a substantial volume of labeled data for effective training. Traditionally, gradient-based (*i.e.*, derivative-based) algorithms are used to obtain the optimal weights and biases of the DNN, *e.g.*, stochastic gradient descent with momentum (SGDM), adaptive momentum estimation (ADAM), and Caputo fractional gradient descent (CFGD) (*Cong, Leung & Wei, 2017*; *Prince, 2023*). Such algorithms, however, suffer from premature convergence and stagnation in local optima, which can degrade the overall performance of the DL models for BC detection.

Recently, some authors have turned to the use of hybrid classification methods that combine ML or DL techniques with global-search optimization algorithms (*e.g.*, metaheuristic) to build optimal models, by attaining the global optimal parameters of the respective model (*Akay, Karaboga & Akay, 2022*). In BC diagnosis, whale optimization algorithm (WOA) and dragonfly algorithm (DA) have been recently employed to obtain the global optimal parameters of SVM (*Elkorany et al., 2022*), which enhances the classification accuracy. DL models have been recently proposed with optimal tuning of the

hyperparameters of the DNN using particle swarm optimization (PSO) (*Aguerchi et al., 2024*), enhanced ant colony optimization (EACO) (*Thirumalaisamy et al., 2023*), gray wolf optimizer (GWO) (*Heikal et al., 2024*), and modified gorilla troops optimization (MGTO) (*Heikal et al., 2024*), or training of the DNN using genetic algorithm (*Davoudi & Thulasiraman, 2021*) and integration of shuffled shepherd optimization (SSO) and deer hunting optimization (DHO) (*Bhausaheb & Kashyap, 2023*). Despite their superiority over gradient-based algorithms in training DNN, metaheuristic algorithms cannot guarantee the global optimal solution due to the need to searching very high dimensional spaces. As is well known, metaheuristics are population-based and adopt random (*i.e.*, gradient-free) search mechanism by utilizing problem-independent operators. Specifically, they start the search by randomly exploring new solutions far from the current ones in the entire search space (exploration phase), and end it up by searching around the best solution obtained so far (exploitation phase). But, searching a high dimensional space without the information about the gradient direction is very challenging. Therefore, metaheuristics still have a tendency to stuck in local optima in training DNN.

To overcome this limitation, researchers have recently attempted to utilize hybrid gradient-metaheuristic optimization algorithms, *e.g.*, cuckoo search algorithm (CSA) with ADAM (*Mohsin, Li & Abdalla, 2020*), beetle antenna search (BAS) with ADAM (*Khan et al., 2020*), and CSA with CFGD (*Habeb et al., 2024*), to train their DL models. Here, gradient-search rules are incorporated into metaheuristic algorithms in order to estimate the gradient direction and thereby direct the search towards the more feasible areas. Such hybrid optimizers have been shown to improve the classification accuracy of the predictive models, where they efficiently search high dimensional spaces, with a smooth convergence behavior and a better escapement from local optima. Although promising, their use in training DL models is still too limited, and further investigation is needed due to the following two reasons. First, further investigation can offer valuable insights for evaluating the applicability of these hybrid optimizers in various scenarios. Second, there is still a requirement for efficient algorithms that can enhance classification accuracy without imposing a substantial computational burden. This research gap is concerning given the massive number of metaheuristic techniques in the literature. In fact, the research field of metaheuristics and their applications in solving optimization problems is highly dynamic since no single metaheuristic algorithm can perform superior in solving all problems according to the No Free Lunch theorem.

The main purpose of this work is to investigate the performance of hybridizing a gradient search mechanism into an efficient metaheuristic technique, multi-verse optimizer (MVO), in training a DL model for BC diagnosis. MVO, which is a nature-inspired optimization technique developed based on cosmology, has shown promising results in various fields such as intelligent computing (*Liu, He & Cui, 2018*; *Kolluru, Gedam & Inamdar, 2018*; *Dif & Elberrichi, 2018*), Internet of Things (IoT) (*Abdel-Basset, Shawky & Eldrandaly, 2020*), signal processing (*Chathoth, Ramdas & Krishnan, 2015*; *Ali et al., 2020*), image segmentation (*Han et al., 2023*), data mining (*Aljarah et al., 2020*), neural networks (*Han et al., 2023*), and software engineering predictive modeling (*Shaheen & El-Sehiemy, 2019*). The motivations behind choosing this algorithm are its good balance

between exploration and exploitation phases and its very few control parameters. To improve the search within the high-dimensional search space of the problem of training DL models, a gradient search rule is incorporated into MVO in order to direct the search towards the more feasible areas by estimating the gradient direction. Unlike previous hybrid optimizers that necessitate considerable computational burden for the adopted gradient search mechanism, the proposed mechanism requires few simple extra computations.

The significant contributions made in this article are manifold.

1. We propose a new optimizer by hybridizing a simple gradient search mechanism with an efficient metaheuristic algorithm, MVO. The proposed hybridization improves the exploration-exploitation balance and convergence speed of original MVO by estimating the gradient direction and directing the search towards the more feasible areas. The new optimizer, which is termed as GMVO, can be effectively used for solving highly non-convex problems with high dimensional search spaces, such as training DNNs.

2. We propose an end-to-end DL model for detecting BC and classifying it as a benign or malignant. In the proposed model, the training of the DNN is carried out using the new hybrid optimizer, GMVO, on WBCD and WDBC. The powerful search capability of GMVO helps in improving the accuracy of BC classification.

3. To evaluate the performance of the proposed method, we train the DNN for twenty times using the proposed optimizer, GMVO, and also using traditional optimizers: SGDM and ADAM, metaheuristic optimizer: MVO, and existing hybrid gradient-metaheuristic optimizers: CSA-ADAM and BAS-ADAM. Based on the outcomes, an average study and statistical analysis have been conducted to demonstrate the superiority of the proposed GMVO-DNN model in terms of classification accuracy and computational efficiency.

The remainder of this article is organized as follows. "Methods and Materials" presents the proposed methodology and the materials. The outcomes are detailed with an in-depth explanation provided in "Results". "Discussions" engages in a comprehensive discussion of the results. The conclusions are then highlighted and the potential avenues for future research are identified.

## METHODS AND MATERIALS

In this section, we outline the methodologies carried out to achieve the research objectives. We start with data description and preprocessing. Next, we introduce the proposed optimizer, GMVO, tailored to enhance the search capability when training DL models. The DNN architecture is then explored, and the block diagram of the proposed GMVO-DNN model for BC diagnosis is presented.

### Data description

In this article, we use two benchmark datasets, namely, WBCD and WDBC, to investigate the performance of the proposed GMVO-DNN model in detecting BC and classifying it as benign or malignant. WBCD and WDBC were compiled by Dr. William H. Wolberg, and

| Table 1 Tumor features in WBCD. | |
|---|---|
| **Attribute** | **Domain** |
| Clump thickness | 1–10 |
| Uniformity of cell size | 1–10 |
| Uniformity of cell shape | 1–10 |
| Marginal adhesion | 1–10 |
| Single epithelial cell size | 1–10 |
| Bare nuclei | 1–10 |
| Bland chromatin | 1–10 |
| Normal nucleoli | 1–10 |
| Mitoses | 1–10 |

obtained from the University of Wisconsin Hospital in Madison. These two datasets are available for academic use through the UCI Machine Learning Repository (https://archive.ics.uci.edu/ml/datasets), where they are given in form of real-valued tumor features with class label. The details of these datasets and their features are given below.

### WBCD

WBCD (original) comprises 699 instances (458 for benign and 241 for malignant) with 11 attributes that represent i) the sample code number, ii) the class label, where '2' denotes benign and '4' denotes malignant, and iii) nine tumor features of fine needle aspirates (FNA) of human breast tissue. The nine tumor features along with their domain of possible values are presented in Table 1. Pathologists assigned numerical values to these nine tumor features based on their observations, with higher values indicating a stronger likelihood of malignancy. Notice that in WBCD, there are 16 instances that have missing values of bare nuclei, which necessitates a robust data prepossessing to avoid accuracy loss.

### WDBC

WDBC comprises 569 instances (357 for benign and 212 for malignant) with 32 attributes that represent i) the ID number of the instance, ii) the class label, where 'N' denotes benign and 'M' denotes malignant, and iii) 30 actual tumor features computed from a digital image of an FNA of a breast mass. The 30 tumor features are resulted from computing mean, standard error, and maximum of ten characteristics of the cell nuclei present in the image. The ten characteristics along with their domain of possible values are given in Table 2. As can be seen, the features have different ranges, which necessitates the normalization of their values in the preprocessing phase. This process reduces the features by taking into consideration the features with impact weight into the classification process and ignoring other features that have not. To have a good classification performance with reduced computational overhead, only 17 features are used, namely, texture_worst, radius_worst, perimeter_worst, perimeter_mean, radius_mean, concave points_worst, concave points_mean, area_worst, area_mean, concavity_mean,

**Table 2 Tumor features in WDBC.**

| Attribute | Domain mean | Standard error | Maximum |
|---|---|---|---|
| Radius | 6.98–28.11 | 0.112–2.873 | 7.93–36.04 |
| Texture | 9.71–39.28 | 0.36–4.89 | 12.02–49.54 |
| Perimeter | 43.79–188.50 | 0.76–21.98 | 50.41–251.20 |
| Area | 143.50–2,501.00 | 6.80–542.20 | 185.20–4,254.00 |
| Smoothness | 0.053–0.163 | 0.002–0.031 | 0.071–0.223 |
| Compactness | 0.019–0.345 | 0.002–0.135 | 0.027–1.058 |
| Concavity | 0.000–0.427 | 0.000–0.396 | 0.000–1.252 |
| Concave points | 0.000–0.201 | 0.000–0.053 | 0.000–0.291 |
| Symmetry | 0.106–0.304 | 0.008–0.079 | 0.157–0.664 |
| Fractal dimension | 0.050–0.097 | 0.001–0.030 | 0.055–0.208 |

concavity_worst, radius_se, area_se, perimeter_se, compactness_mean, compactness_worst, texture_mean.

## Data preprocessing

A raw dataset usually suffers from some data issues, such as unwanted noise, categorical data that cannot be handled by models, redundant data, and missing data, which can significantly impact the accuracy and reliability of the subsequent analyses. Therefore, data preprocessing is an important step to convert the raw dataset into a clean dataset consisting of well-formed data that can be fed into classification models. In this work, we effectively preprocessed the two raw datasets, WBCD and WDBC, and prepared them to ensure their potential suitability for training and testing processes. The proposed data preprocessing steps include cleaning, encoding, normalization, and splitting, which are described as follows.

We started by thoroughly cleaning the raw dataset to reduce the unwanted noise, remove any redundant data, and handle the missing data. For example, the sample code number in WBCD and the ID number in WDBC, were removed since they are irrelevant to the diagnosis. We replaced the mean of the numerical distribution with the 16 missing values of the 'bare nuclei' attribute in WBCD. The next step is the application of one-hot encoding to convert class vectors into a binary class matrix. The label encoding was also employed to transform the categorical data into numerical format that can be handled by the classification model. Particularly, the 'Class' attribute in WBCD and 'Diagnosis' attribute in WDBC were transformed into 0 and 1, where 1 corresponds to a malignant tumor and 0 corresponds to a benign tumor. Following these steps, data normalization was carried out to restrict the values of the tumor features within a consistent range of 0 to 1. This step not only speeds up the model learning process but also helps in overcoming overfitting and underfitting problems.

The data was split into a training set comprising 70% of the data and a testing set consisting of the remaining 30%. During the evaluation process, a 10-fold cross-validation method was employed to ensure reliable results and to prevent the overfitting problem.

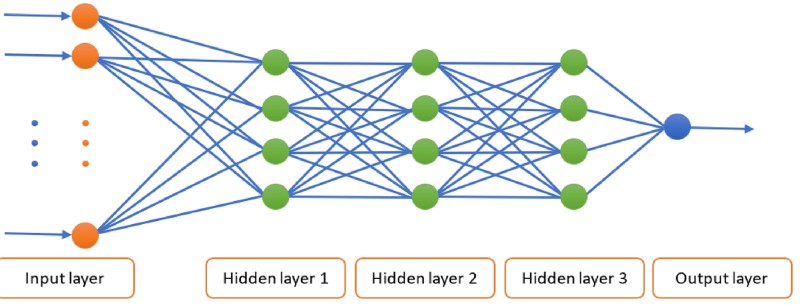

**Figure 1 Block diagram of optimizing the parameters of the DNN using MVO.**

This method involves randomly splitting the training set into 10 folds of equal size. By running ten iterations, each fold is used as the validation set once, while the remaining nine folds serve as the training set. The average of the results obtained in the ten runs is considered as the validation result of the model.

## DNN architecture

The proposed DL model for BC diagnosis utilizes the DNN architecture shown in Fig. 1. This DNN is composed of three hidden layers with an input layer and an output layer. The use of three hidden layers is driven by the need to strike a balance between creating a sufficiently complex model that can capture intricate details and keeping it simple enough to perform effectively. The number of nodes in the input layer is equal to the number of features in the respective training dataset. That is, 9 and 17 nodes are used with WBCD and WDBC respectively. The numbers of nodes in the hidden layers are, respectively, 5, 3, and 2 for WBCD and 8, 4, and 2 for WDBC. To expedite learning and promote faster convergence, the ReLU activation function is applied in each hidden layer. In the output layer, a single output node is used for binary representation of the diagnosis outcomes, where the Sigmoid activation function is applied.

## The proposed GMVO optimizer

For a real-valued function $f(\mathbf{x})$, $\mathbf{x} = [x_0, x_1, \cdots, x_d] \in \mathbb{R}^d$, it is often required to find the value of $\mathbf{x}$ that minimize $f(\mathbf{x})$, *i.e.*, $\min_{\mathbf{x} \in \mathbb{R}^d} f(\mathbf{x})$. The conventional gradient decent method has the following iterative rule for $\mathbf{x}$:

$$\mathbf{x}^{(k+1)} = \mathbf{x}^{(k)} - \eta_k \cdot \nabla_{\mathbf{x}} f\left(\mathbf{x}^{(k)}\right), \tag{1}$$

where $k$ denotes the number of iteration and $\eta_k$ is the learning rate at the $k$th iteration. Such a method conducts the search for an optimal solution by taking discrete steps in the direction of steepest descent. It has been used to solve many optimization problems, *e.g.*, training of neural networks. Often, this method converges linearly to a stationary point provided that the learning rates are appropriately chosen. In DL models for BC diagnosis, the gradient-based methods, SGDM and ADAM, have been widely used in adjusting the

model's weights during training. However, these methods converge to a local minimum point (*Li et al., 2012*; *Belciug & Gorunescu, 2013*).

Global search techniques (*e.g.*, metaheuristics) have been proposed for optimizing the weights of the neural network in order to enhance the overall performance of the predictive models (*Davoudi & Thulasiraman, 2021*; *Bhausaheb & Kashyap, 2023*). However, such techniques still have a tendency to stuck in local optima due to the need to searching very high dimensional spaces without the information about the gradient direction.

Recently, the training of DL models have been proposed using hybrid optimizers (*Mohsin, Li & Abdalla, 2020*; *Khan et al., 2020*; *Habeb et al., 2024*), which incorporate gradient-search rules into metaheuristic algorithms in order to estimate the gradient direction and thereby direct the search towards the more feasible areas. Such hybrid optimizers have been shown to improve the classification accuracy of the predictive models, where they efficiently search high dimensional spaces with a smooth convergence behavior and a better escapement from local optima.

In this work, we incorporate a simple gradient search mechanism to a recently-proposed metaheuristic technique, MVO, in order to device a hybrid optimizer, named GMVO, with improved global search capabilities.

### Multi-verse optimizer

MVO is a nature-inspired optimization technique that utilizes mathematical models of the three concepts of multi-verse theory: white holes, black holes, and wormholes (*Mirjalili, Mirjalili & Hatamlou, 2016*). White holes are considered as the source of universes creation. Black holes are associated with exceptional gravitational forces, and hence can captivate any object. Wormholes enable instant movement of objects within a universe or between universes. In MVO, the exploration of the search space is carried out based on white and black holes, while the exploitation around the best solution is boosted using wormholes. The population size corresponds to the number of universes, where each universe representing a solution and its objects representing variables. An inflation rate is assigned to each universe. This inflation rate is proportional to fitness value of the corresponding solution. MVO follows specific rules during optimization to guide the search for the optimal solution. These rules are:

1. The presence of a white hole is directly proportional to the inflation rate. In contrast, the presence of a black hole is inversely proportional to the inflation rate.
2. Universes with larger inflation rates tend to transmit objects *via* white holes. On the other hand, universes with smaller inflation rates tend to captivate objects *via* black holes.
3. The objects within all universes, regardless their inflation rates, have the potential to travel in a random manner towards the best universe *via* wormholes.

During optimization, the objects are transferred through white/black hole tunnels, namely, from the white holes of the universe that has a larger inflation rate to the black holes of the universe that has a smaller inflation rate. In this way, the average inflation rate of the entire universes is improved with the optimization progress. A sorting of the

universes according to the inflation rate is performed in each iteration, and a roulette wheel selection is used to pick a universe to possess a white hole.

Let $n$ be the population size (*i.e.*, the number of the universes) and $d$ is the dimensions of each candidate solution (*i.e.*, the number of objects in each universe). Then, the universes are modeled by

$$\mathbf{X} = \begin{bmatrix} x_{11} & \cdots & x_{1d} \\ \vdots & \ddots & \vdots \\ x_{n1} & \cdots & x_{nd} \end{bmatrix}. \tag{2}$$

The exchange of objects between universes is carried out using the following equation:

$$x_{ij} = \begin{cases} x_{kj}, & r_1 < \tilde{F}_i \\ x_{ij}, & r_1 \geq \tilde{F}_i, \end{cases} \tag{3}$$

where $x_{ij}$ and $x_{kj}$ denote the $j$th variable of the $i$th universe and the $k$th universe selected by the roulette wheel selection, respectively. $r_1 \in [0, 1]$ denotes a random number, $\mathbf{x}_i$ denotes the $i$th universe, and $\tilde{F}_i$ denotes normalized inflation rate of the $i$th universe. The mechanism of exchanging objects between universes results in abrupt changes of universes, thereby ensures the exploration of the search space with avoidance of local optima stagnation.

To carry out exploitation, wormholes are used to transfer objects randomly regardless the inflation rates of the respective universes. Wormhole tunnels are regularly formed between a universe and the best universe obtained so far to emphasize local changes for each universe. This also increases the possibility of enhancing the inflation rates of all universes. This mechanism is formulated as

$$x_{ij} = \begin{cases} \begin{cases} \mathbf{x}_{best}(j) + v_T \times ((ub_j - lb_j) \times r_4 + lb_j) & r_3 < 0.5 \\ \mathbf{x}_{best}(j) - v_T \times ((ub_j - lb_j) \times r_4 + lb_j) & r_3 \geq 0.5 \end{cases} & r_2 < P_w \\ x_{ij} & r_2 \geq P_w \end{cases}, \tag{4}$$

where $x_{ij}$ denotes the $j$th variable of the $i$th universe, and $\bar{x}_j$ is the $j$th variable of the best universe obtained so far. $ub_j$ and $lb_j$ are the upper and lower bounds of variables, respectively, and $r_2$, $r_3$, and $r_4$ are random numbers in $[0, 1]$. $P_w$ indicates the existence probability of wormholes in universes, where it is computed over iterations by

$$P_w = P_{min} + m \times \left( \frac{P_{max} - P_{min}}{M} \right) \tag{5}$$

where $P_{min}$ and $P_{max}$ are the lower and upper bounds of $P_w$, respectively (commonly $P_{min} = 0.2$ and $P_{max} = 1$). $m$ is the iteration number, and $M$ is the maximum number of iterations. $v_T$ indicates the traveling distance rate across which an object travels *via* a wormhole around the best universe obtained so far. It is iteratively updated as follows

$$v_T = 1 - \left( \frac{m}{M} \right)^{1/\chi}, \tag{6}$$

where $\chi$ is a parameter used to specify the exploitation accuracy over the iterations (often $\chi$

is set to 6). As the optimization progresses, $P_w$ increases linearly to emphasize exploitation, meanwhile $v_T$ decreases in order to ensure precise local search around the best universe formed so far. The use of these two parameters guarantees the convergence of MVO towards an optimal solution and promotes a good balance between exploration and exploitation phases. MVO has shown promising results in various fields such as intelligent computing (*Liu, He & Cui, 2018*; *Kolluru, Gedam & Inamdar, 2018*; *Dif & Elberrichi, 2018*), IoT (*Abdel-Basset, Shawky & Eldrandaly, 2020*), signal processing (*Chathoth, Ramdas & Krishnan, 2015*; *Ali et al., 2020*), image segmentation (*Han et al., 2023*), data mining (*Aljarah et al., 2020*), neural networks (*Han et al., 2023*), and software engineering predictive modeling (*Shaheen & El-Sehiemy, 2019*). However, like any other metaheuristic, MVO has a tendency to stuck in local optima in training DNN, due to the need to searching very high dimensional spaces without the information about the gradient direction.

### Integration of gradient rule with MVO

The proposed algorithm, GMVO, integrates a gradient search rule into MVO in order to improve the capability of solving optimization problems with high dimensional spaces such as training of DL models. In GMVO, the traveling distance rate across which an object transports by a wormhole is adapted based on derivative of the objective function of the optimization problem. This adaptation aims to allocate more resources to regions with significant changes in the objective function, facilitating effective exploration and exploitation of the solution space.

To model the new traveling distance rate mathematically, we introduce the following equation:

$$\bar{v}_T = v_T \cdot S, \tag{7}$$

where $S$ is a derivative-based scaling factor and $v_T$ is given in Eq. (6). Note that the traveling distance rate existed in conventional MVO, $v_T$, is maintained to guarantee the convergence towards an optimal solution over the course of iterations. Yet, we incorporate $S$ to adjust the traveling distance rate based on the derivative of the objective function. This factor is given by

$$S = 1 + \alpha \cdot f'(\mathbf{x}), \tag{8}$$

where $f'(\mathbf{x})$ denotes the derivative of the objective function with respect to $\mathbf{x}$. The parameter $\alpha$ is used to control the ratio of the derivative(gradient)-based component to the conventional component of the traveling distance rate. Obviously, if $\alpha \gg 1$, then the effect of the gradient-based component of $\bar{v}_T$ dominates over the conventional one $v_T$. On the other hand, if $\alpha \ll 1$, the the conventional component $v_T$ dominates, and GMVO is reduced to MVO as $\alpha \to 0$.

Since most of optimization problems are non-differentiable, $f'(\mathbf{x})$ is computed using a numerical gradient technique. Using the truncated Taylor series, $f'(\mathbf{x})$ can be approximated by the following central differencing formula:

---

**Algorithm 1** GMVO.

---

**begin**

Objective function $f(\mathbf{x}), \mathbf{x} = (x_1, \ldots, x_d)^T$

Generate initial population of $n$ universes $\mathbf{x}_i \, (i = 0, 1, \cdots, n)$

Initialize $P_w, v_T, \bar{v}_T, \mathbf{x}_{best}$, and $\mathbf{x}_{worst}$

$\overline{\mathbf{X}} =$ Sorted universes

$\widetilde{\mathbf{F}} =$ normalize the fitnesses of the universes

**while** the end criterion is not satisfied **do**

    Evaluate the objective function $f(\mathbf{x}_i)$ for $i = 0, 1, \cdots, n$

    **for** each universe $\mathbf{x}_i$ **do**

        Update $P_w$

        Update $v_T$ and $\bar{v}_T$

        *Black_hole_index = i;*

        **for** each object indexed by $j$ **do**

            $r_1 = random([0, 1]);$

            **if** $r_1 < \widetilde{\mathbf{F}}_i$ **then**

                *White_hole_index = RouletteWheelSelection$(-\widetilde{\mathbf{F}})$;*

                $\mathbf{X}(Black\_hole\_index, j) = \overline{\mathbf{X}}(White\_hole\_index, j);$

            **end if**

            $r_2 = random([0, 1]);$

            **if** $r_2 < P_w$ **then**

                $r_3 = random([0, 1]);$

                $x_{ij} = \mathbf{x}_{best}(j) + \bar{v}_T \times ((ub(j) - lb(j)) \times r_3 + lb(j));$

            **end if**

        **end for**

    **end for**

**end while**

**end**

---

$$f'(\mathbf{x}) = \frac{f(\mathbf{x} + \Delta\mathbf{x}) - f(\mathbf{x} - \Delta\mathbf{x})}{2\Delta\mathbf{x}}. \tag{9}$$

In this study, we substitute the position $\mathbf{x} - \Delta\mathbf{x}$ with $\mathbf{x}_{best}$ and the position $\mathbf{x} + \Delta\mathbf{x}$ with $\mathbf{x}_{worst}$. The solutions $\mathbf{x}_{best}$ and $\mathbf{x}_{worst}$ correspond to a better and a worse fitness in the vicinity of position $\mathbf{x}$ respectively. To reduce the computational burden per iteration, we use the position $\mathbf{x}$ rather than $f(\mathbf{x})$. To ensure a different step for each candidate solution and increase the diversity of population around $\mathbf{x}_{best}$, $\Delta\mathbf{x}$ is randomly selected from the interval $[0, 0.001]$. The high diversity of population promotes exploration with local optima avoidance. With the mentioned substitutions, $S$ can be expressed by

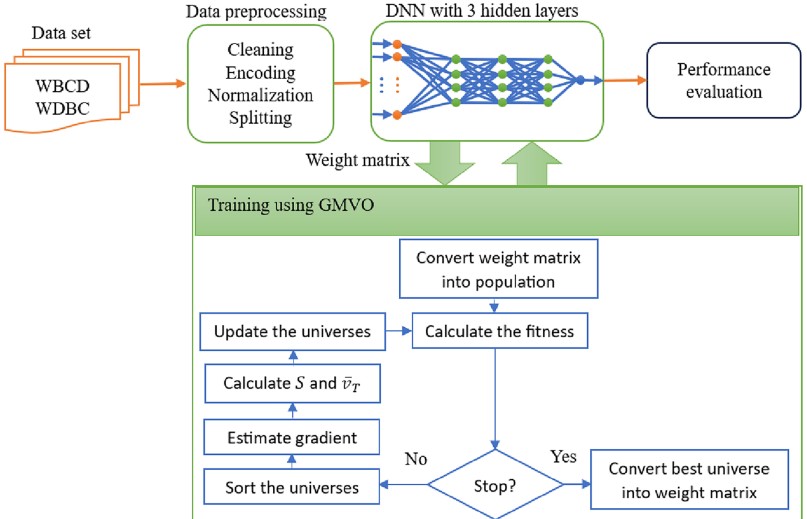

**Figure 2** The traveling distance rates during iterations (A) before the adjustment, (B) after the adjustment.           

$$S = 1 + \alpha \cdot \frac{\mathbf{x}_{best} - \mathbf{x}_{worst} + \varepsilon}{2\Delta\mathbf{x}}, \tag{10}$$

where $\varepsilon$ is a small number. Note that the objective function has its minimum and maximum values around $\mathbf{x}$ at the positions $\mathbf{x}_{best}$ and $\mathbf{x}_{worst}$ respectively. Therefore, the vector $\mathbf{x}_{best} - \mathbf{x}_{worst}$ represents the gradient direction. Substituting Eq. (10) into Eq. (7) yields

$$\bar{v}_T = v_T \left(1 + \alpha \cdot \frac{\mathbf{x}_{best} - \mathbf{x}_{worst} + \varepsilon}{2\Delta\mathbf{x}}\right). \tag{11}$$

Thus, the update rule for $x_{ij}$ in GMVO is given by

$$x_{ij} = \begin{cases} \mathbf{x}_{best}(j) + \bar{v}_T \times ((ub_j - lb_j) \times r_4 + lb_j) & r_2 < \mathrm{P}_w \\ x_{ij} & r_2 \geq \mathrm{P}_w \end{cases}. \tag{12}$$

The adjusted traveling distance rate value, $\bar{v}_T$, reflects the influence of the gradient on the step size of the search process. A higher gradient value results in a larger $\bar{v}_T$, allowing for more extensive exploration and exploitation in regions with significant changes in the objective function. Conversely, a lower gradient value results in a smaller $\bar{v}_T$, enabling finer-grained exploration and exploitation in regions with smoother objective function landscapes. Moreover, it can be inferred from Eq. (12) and Eq. (11) that the proposed algorithm does not require much extra computations compared to original MVO, where the vectors $\mathbf{x}_{best}$ and $\mathbf{x}_{worst}$ can be easily acquired from the sorted universe matrix.

The pseudocode for GMVO is given below.

## The model framework

The block diagram of the proposed DL model for BC diagnosis is shown in Fig. 2. The proposed optimizer, GMVO, is used to refine the weights of the DNN as illustrated in

**Algorithm 2** Optimizing DNN through GMVO.

**begin**

Initialize the DNN architecture;

Initialize random weights for the DNN;

Create a matrix for the weights;

Initialize the population of GMVO;

**while** Termination condition is not true **do**

    Predict the output of DNN for the weight matrix;

    Calculate the objective function for all universes;

    Sort the universes based on the fitness value;

    Estimate the gradient direction $\mathbf{x}_{best} - \mathbf{x}_{worst}$;

    Update the universes;

    Update the weight matrix with $\mathbf{x}_{best}$;

**end while**

**end**

Algorithm 2. The initial weights are stored in a weight matrix, which is then converted into a vector that serves as the initial population for GMVO. The parameters for GMVO are set to $M = 1{,}000$, $P_{min} = 0.5$, $P_{max} = 1$, $\chi = 6$, and $\alpha = 0.7$. Note that $P_{min} = 0.5$ (instead of 0.2 in original MVO) increases the probability of earlier gradient-based search in the large dimensional space of our optimization problem. In the same time, $\alpha = 0.7$ gives more importance for the gradient-based component of the traveling distance rate when updating the universes.

## Performance evaluation metrics

The performance of the proposed GMVO-DNN model is evaluated on the two widely-known datasets WBCD and WDBC. The classification outcomes are categorized into true positive (TP) indicating cases that correctly diagnosed as malignant, true negative (TN) indicating cases that correctly diagnosed as malignant, false positive (FP) indicating cases that wrongly diagnosed as malignant, and false negative (FN) indicating cases that wrongly diagnosed as benign. Based on these four categories, the following performance metric are computed:

$$\text{Accuracy (ACC)} = \frac{TP + TN}{TP + TN + FP + FN}, \tag{13}$$

$$\text{Precision (positive predictive value, PPV)} = \frac{TP}{TP + FP}, \tag{14}$$

$$\text{Specificity (SPC)} = \frac{TN}{TN + FP}, \tag{15}$$

$$\text{Sensitivity (SEN)} = \frac{TP}{TP + FN}, \tag{16}$$

$$\text{F1 } Score = \frac{2 \cdot \text{Precision} \cdot \text{Sensitivity}}{\text{Precision} + \text{Sensitivity}}, \tag{17}$$

and

$$\text{Matthew correlation coefficient (MCC)} = \frac{TP \cdot TN - FP \cdot FN}{\sqrt{(TP + FP)(TP + FN)(TN + FP)(TN + FN)}}. \tag{18}$$

To validate the effectiveness of our proposed optimizer, we compare its performance with that of original MVO, and the well-known optimizers, *i.e.*, SGDM and ADAM, as well as hybrid recently-proposed optimizers, *i.e.*, BAS-ADAM (*Khan et al., 2020*), and CSA-ADAM (*Mohsin, Li & Abdalla, 2020*). That is, each optimizer was employed to train the DNN using WBCD and WDBC. The experiment for each optimizer was repeated for twenty times, and the above performance metrics are computed.

## RESULTS

In this section, the results of the conducted experiments are presented. The parameters used in the experiments were, population size of 50, learning rate of $10^{-3}$, batch size of 10, and number of epochs of 6. All the experiments are executed on Intel Core i7-1260P CPU processor with a 64-bit Windows 11 operating system and 8.00-GB RAM using MATLAB.

Figure 3 (left panel) illustrates the accuracy for each run of the twenty runs for the optimizers being compared, whereas Fig. 3 (right panel) illustrates the accuracy *vs.* epoch curve for the best run of each optimizer. The top figures are for WBCD and the bottom figures are for WDBC. As can be clearly seen, the accuracy of the MVO-based model is similar or slightly better as compared to those of the SGDM and ADAM-based models. However, the proposed GMVO-based model achieves accuracy significantly better than that of all the other models, which emphasizes the benefit gained from incorporating the gradient search rule into MVO.

For further illustration, the overall performance over the twenty runs for each optimizer is investigated in terms of mean, best, and worst accuracy along with standard deviation and average computational time. The results are presented in Table 3. From this table, it is evident that the proposed model steadily outperforms all the others, where it achieves average accuracy of 93.95% and 96.73% for WBCD and WDBC respectively. The smallest standard deviation values of our model indicate its stable performance and robustness.

Moreover, statistical hypothesis test is conducted to compare the consistency in the performance of our model with that of the models based on SGDM, ADAM, CSA-ADAM, BAS-ADAM, and MVO. Namely, Table 4 presents the results of Wilcoxon rank-sum test over the twenty runs for the accuracy metric. Here, $n_1$ and $n_2$ denote the number of outcomes considered from those of SGDM, ADAM, CSA-ADAM, BAS-ADAM, or MVO-based model and those of our model, respectively. $W_1$ and $W_2$ denote the respective sum of ranks. The null hypothesis is either rejected at a level of significance $a$ or accepted based on a comparison between $W_1$ and $W_2$ with the critical values shown in Table 4. The

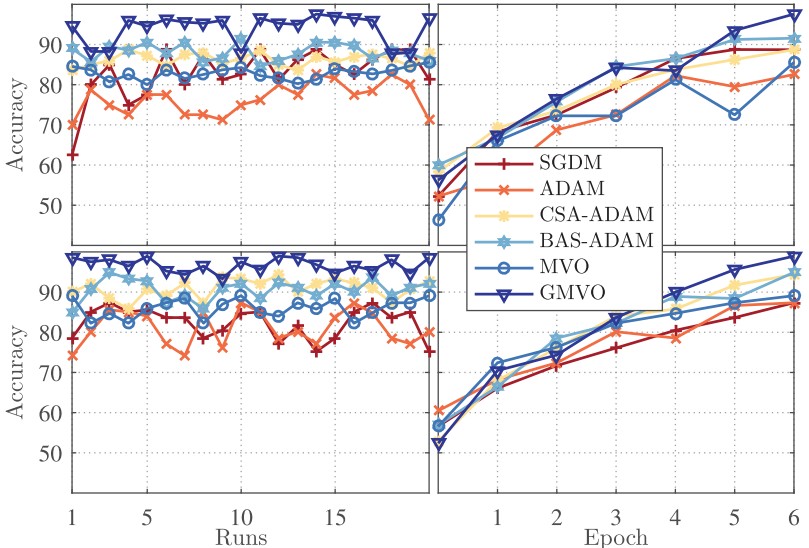

**Figure 3 The confusion matrices of each algorithm (A) SGDM, (B) ADAM, (C) MVO, (D) CSA-ADAM, (E) BAS-ADAM AND (F) GMVO.**

**Table 3 Overall performance over the twenty runs in terms of accuracy.**

| | WBCD | | | | | WDBC | | | | |
|---|---|---|---|---|---|---|---|---|---|---|
| | Accuracy | | | | | Accuracy | | | | |
| Optimizer | Mean | Best | Worst | Std | $T_{avg}$ | Mean | Best | Worst | Std | $T_{avg}$ |
| SGDM | 82.83 | 88.70 | 62.52 | 6.30 | 230 | 82.33 | 87.35 | 75.22 | 3.86 | 376 |
| ADAM | 76.68 | 82.55 | 69.96 | 3.88 | 240 | 81.06 | 87.35 | 74.17 | 4.33 | 382 |
| CSA-ADAM | 86.55 | 88.70 | 83.55 | 1.58 | 277 | 90.93 | 94.38 | 85.76 | 2.36 | 409 |
| BAS-ADAM | 88.25 | 91.56 | 84.55 | 2.05 | 289 | 90.56 | 94.90 | 84.89 | 2.61 | 418 |
| MVO | 82.83 | 85.55 | 79.97 | 3.58 | 156 | 85.98 | 89.10 | 82.25 | 2.44 | 345 |
| GMVO | 93.95 | 97.57 | 87.70 | 1.55 | 180 | 96.73 | 98.95 | 93.32 | 1.71 | 362 |

confidence level (CL) is computed by $CL = (1 − a) \times 100\%$. As can be seen from this table, the null hypothesis is rejected with 99% and 95% level of confidence in all cases except one case. Therefore, the proposed model can be regarded as the most accurate one in BC diagnosis using WBCD and WDBC datasets.

To obtain a thorough evaluation of performance, it is crucial to take into account metrics beyond accuracy. Table 5 presents the average values of the performance metrics, ACC, PPV, SPC, SEN, F1 score, and MCC over the twenty runs of each optimizer. Notice that PPV refers to the fraction of correctly detected malignant patients out of all of the predicted malignant patients, SPC refers to the model's ability to correctly identify healthy (benign) patients, and SEN refers to the model's ability to correctly detect malignant patients out of the actual malignant patients. As can be seen from Table 5, our model achieves the highest PPV, SPC, and SEN values for both WBCD and WDBC compared to the other models. This indicates that our model has smallest number of falsely predicted

**Table 4 Wilcoxon rank-sum test over the twenty runs.**

| | | | Wilcoxon rank-sum test | | | | | | | | |
|---|---|---|---|---|---|---|---|---|---|---|---|
| | | | Critical value | | WBCD | | | WDBC | | | |
| Optimizer | $n_1$ | $n_2$ | 0.05 | 0.01 | $W_1$ | $W_2$ | Accept/Reject (CL) | $W_1$ | $W_2$ | Accept/Reject (CL) |
| SGDM | 7 | 11 | 44 | 38 | 33 | 138 | Reject (99%) | 31 | 140 | Reject (99%) |
| | 10 | 12 | 85 | 76 | 61 | 192 | Reject (99%) | 60 | 193 | Reject (99%) |
| ADAM | 7 | 11 | 44 | 38 | 28 | 143 | Reject (99%) | 28 | 143 | Reject (99%) |
| | 10 | 12 | 85 | 76 | 55 | 198 | Reject (99%) | 60 | 193 | Reject (99%) |
| CSA-ADAM | 7 | 11 | 44 | 38 | 37 | 134 | Reject (99%) | 40 | 131 | Reject (95%) |
| | 10 | 12 | 85 | 76 | 71 | 182 | Reject (99%) | 79 | 174 | Reject (95%) |
| BAS-ADAM | 7 | 11 | 44 | 38 | 43 | 128 | Reject (95%) | 40 | 131 | Reject (95%) |
| | 10 | 12 | 85 | 76 | 88 | 165 | Accept | 79 | 174 | Reject (95%) |
| MVO | 7 | 11 | 44 | 38 | 33 | 138 | Reject (99%) | 35 | 136 | Reject (99%) |
| | 10 | 12 | 85 | 76 | 64 | 189 | Reject (99%) | 71 | 182 | Reject (99%) |

**Table 5 Average values of various performance metrics over the twenty runs.**

| Optimizer | WBCD | | | | | | WDBC | | | | | |
|---|---|---|---|---|---|---|---|---|---|---|---|---|
| | ACC | PPV | SPC | SEN | F1 Score | MCC | ACC | PPV | SPC | SEN | F1 Score | MCC |
| SGDM | 82.83 | 72.06 | 82.72 | 83.15 | 77.16 | 64.05 | 82.33 | 76.40 | 86.19 | 75.83 | 76.09 | 62.10 |
| ADAM | 76.68 | 63.09 | 75.93 | 77.61 | 69.58 | 51.54 | 81.06 | 74.89 | 85.46 | 73.66 | 74.24 | 59.30 |
| CSA-ADAM | 86.55 | 76.38 | 85.74 | 87.34 | 81.48 | 71.10 | 90.93 | 85.33 | 90.71 | 91.30 | 88.20 | 80.99 |
| BAS-ADAM | 88.25 | 78.91 | 87.36 | 89.69 | 83.94 | 75.03 | 90.56 | 85.02 | 90.56 | 90.57 | 87.70 | 80.17 |
| MVO | 82.83 | 71.42 | 82.34 | 83.71 | 77.06 | 63.99 | 85.98 | 80.50 | 88.21 | 82.24 | 81.35 | 70.15 |
| GMVO | 93.95 | 88.06 | 93.06 | 95.64 | 91.67 | 87.14 | 96.73 | 93.38 | 95.83 | 98.25 | 95.75 | 93.18 |

and undetected malignant patients. Since WBCD and WDBC are imbalanced, the F1 score is an effective evaluation metric as a measure of the harmonic mean of PPV and SEN. MCC is even more informative by considering the balance ratios of TPs, TNs, FPs, and FNs. From the results, both F1 score and MCC of the proposed model are much higher than those of the other models. To offer a detailed breakdown of the models' predictive capabilities, Fig. 4 depicts confusion matrix for the best run of each optimizer for WBCD. This figure confirms that our approach is superior to the other ones in terms of BC diagnosis.

The scalability of classification model is vital in practical scenarios with the volume of data continuously grows. That is, preserving high accuracy with larger volume of data emphasizes the capability of the model to handle ever-growing and diversified datasets. Table 6 presents performance evaluation of the competing models on different fractions of datasets. (Notice that the results for the 100% data size are those of the best run over the twenty runs of each model.) From this table it can be inferred that, in general, the evaluation metrics for all models are consistently improved with increasing the data size.

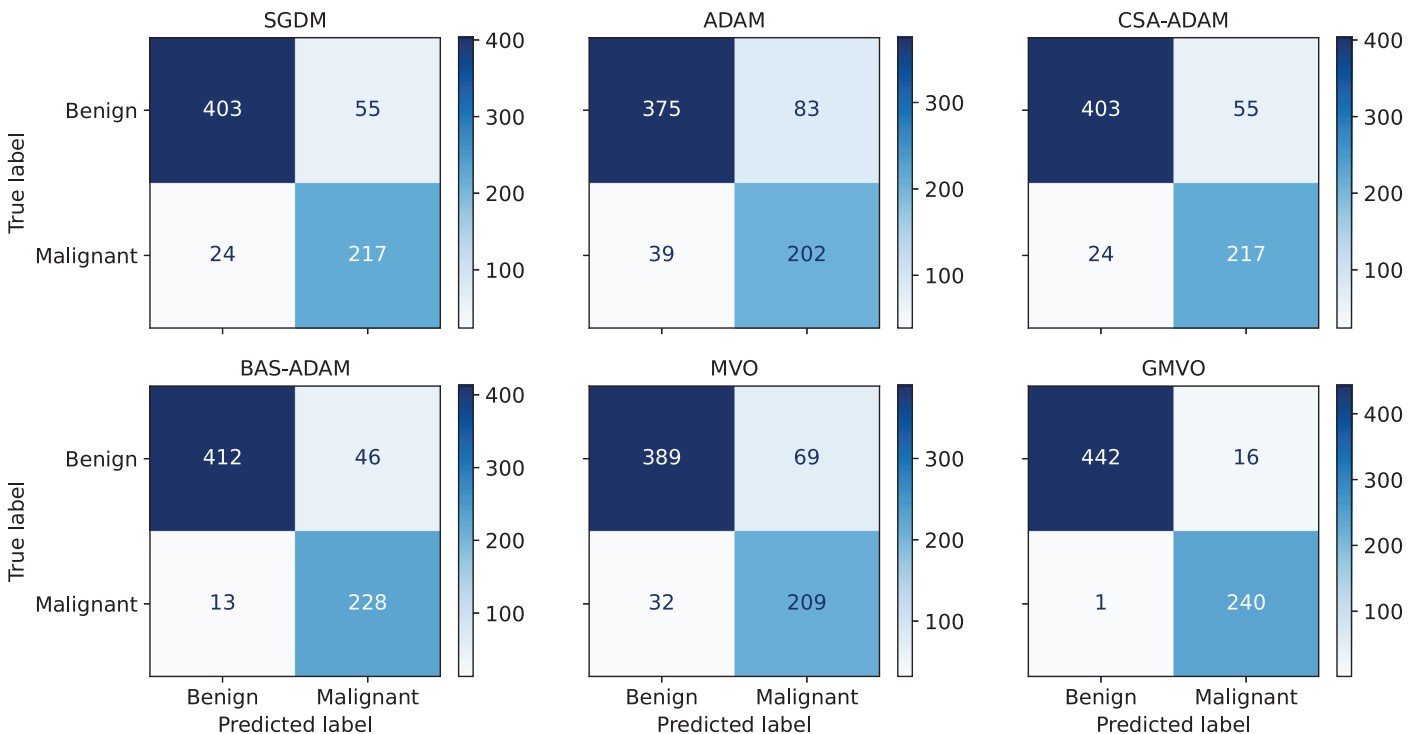

**Figure 4 Block diagram of optimizing the parameters of the DNN using MVO.** The average accuracy curves of the optimizers used in the experiments (A) CFGD and (B) CSA-CFGD with different values of α.               

**Table 6 Performance metrics for the competing models on different fractions of datasets.**

| Optimizer | Data size | WBCD | | | | | | WDBC | | | | | |
|---|---|---|---|---|---|---|---|---|---|---|---|---|---|
| | | ACC | PPV | SPC | SEN | F1 Score | MCC | ACC | PPV | SPC | SEN | F1 Score | MCC |
| SGDM | 25% | 84.00 | 78.57 | 89.57 | 73.33 | 75.86 | 64.00 | 79.72 | 70.69 | 81.11 | 77.36 | 73.87 | 57.51 |
| | 50% | 87.71 | 79.55 | 88.21 | 86.78 | 83.00 | 73.58 | 81.40 | 74.77 | 84.92 | 75.47 | 75.12 | 60.27 |
| | 75% | 89.71 | 83.60 | 90.99 | 87.29 | 85.41 | 77.51 | 84.07 | 75.71 | 83.96 | 84.28 | 79.76 | 66.96 |
| | 100% | 88.70 | 79.78 | 87.99 | 90.04 | 84.60 | 76.07 | 87.35 | 82.11 | 89.08 | 84.43 | 83.26 | 73.11 |
| ADAM | 25% | 74.86 | 63.79 | 81.74 | 61.67 | 62.71 | 43.77 | 75.52 | 65.52 | 77.78 | 71.70 | 68.47 | 48.67 |
| | 50% | 83.71 | 72.86 | 83.41 | 84.30 | 78.16 | 65.73 | 79.65 | 72.64 | 83.80 | 72.64 | 72.64 | 56.44 |
| | 75% | 86.10 | 80.34 | 89.83 | 79.01 | 79.67 | 69.11 | 81.26 | 71.58 | 80.60 | 82.39 | 76.61 | 61.53 |
| | 100% | 82.55 | 70.88 | 81.88 | 83.82 | 76.81 | 63.54 | 87.35 | 82.11 | 89.08 | 84.43 | 83.26 | 73.11 |
| CSA-ADAM | 25% | 90.86 | 85.48 | 92.17 | 88.33 | 86.89 | 79.90 | 87.41 | 79.66 | 86.67 | 88.68 | 83.93 | 73.92 |
| | 50% | 91.43 | 86.40 | 92.58 | 89.26 | 87.80 | 81.23 | 90.18 | 86.11 | 91.62 | 87.74 | 86.92 | 79.06 |
| | 75% | 92.95 | 89.56 | 94.48 | 90.06 | 89.81 | 84.42 | 90.63 | 85.63 | 91.04 | 89.94 | 87.73 | 80.22 |
| | 100% | 88.70 | 79.78 | 87.99 | 90.04 | 84.60 | 76.07 | 94.38 | 89.13 | 93.00 | 96.70 | 92.76 | 88.37 |
| BAS-ADAM | 25% | 92.00 | 88.33 | 93.91 | 88.33 | 88.33 | 82.25 | 83.22 | 75.44 | 84.44 | 81.13 | 78.18 | 64.69 |
| | 50% | 93.14 | 88.80 | 93.89 | 91.74 | 90.24 | 84.99 | 85.96 | 82.35 | 89.94 | 79.25 | 80.77 | 69.76 |
| | 75% | 93.33 | 90.11 | 94.77 | 90.61 | 90.36 | 85.26 | 86.65 | 79.65 | 86.94 | 86.16 | 82.78 | 72.06 |
| | 100% | 91.56 | 83.21 | 89.96 | 94.61 | 88.54 | 82.33 | 94.90 | 90.31 | 93.84 | 96.70 | 93.39 | 89.39 |

| Table 6 (continued) | | | | | | | | | | | | | |
| Optimizer | Data size | WBCD | | | | | | WDBC | | | | | |
| | | ACC | PPV | SPC | SEN | F1 Score | MCC | ACC | PPV | SPC | SEN | F1 Score | MCC |
| MVO | 25% | 81.71 | 75.00 | 87.83 | 70.00 | 72.41 | 58.84 | 77.62 | 67.80 | 78.89 | 75.47 | 71.43 | 53.33 |
| | 50% | 88.86 | 81.06 | 89.08 | 88.43 | 84.58 | 76.06 | 82.11 | 75.70 | 85.47 | 76.42 | 76.06 | 61.77 |
| | 75% | 90.86 | 87.15 | 93.31 | 86.19 | 86.67 | 79.71 | 82.67 | 73.74 | 82.46 | 83.02 | 78.11 | 64.16 |
| | 100% | 85.55 | 75.18 | 84.93 | 86.72 | 80.54 | 69.59 | 89.10 | 83.78 | 89.92 | 87.74 | 85.71 | 76.97 |
| GMVO | 25% | 96.57 | 96.55 | 98.26 | 93.33 | 94.92 | 92.36 | 94.41 | 90.91 | 94.44 | 94.34 | 92.59 | 88.14 |
| | 50% | 97.14 | 95.87 | 97.82 | 95.87 | 95.87 | 93.68 | 97.54 | 97.14 | 98.32 | 96.23 | 96.68 | 94.74 |
| | 75% | 97.90 | 96.20 | 97.97 | 97.79 | 96.99 | 95.39 | 98.83 | 98.73 | 99.25 | 98.11 | 98.42 | 97.49 |
| | 100% | 97.57 | 93.75 | 96.51 | 99.59 | 96.58 | 94.80 | 98.95 | 97.25 | 98.32 | 100.00 | 98.60 | 97.78 |

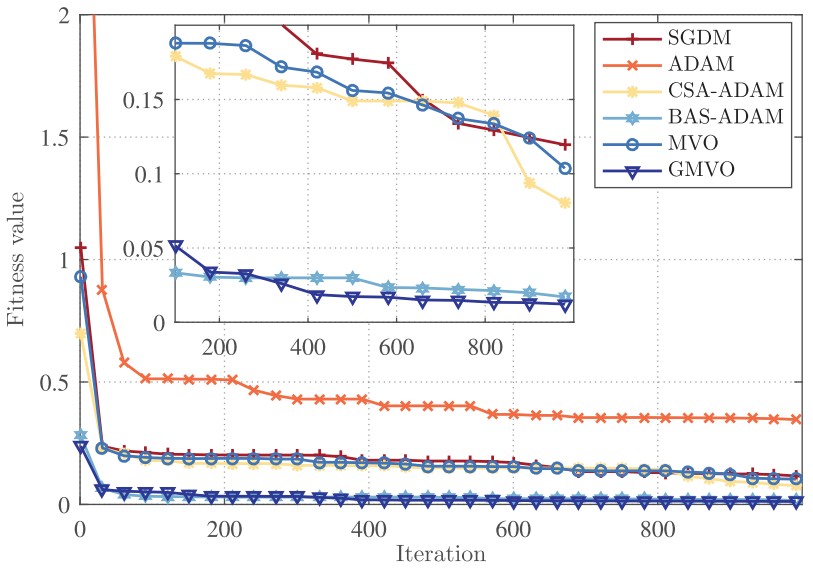

**Figure 5 Convergence profiles for an arbitrary iteration.**

An exception to this general tendency is that, some performance setback for all models in terms of most of the evaluation metrics is observed as the data size is increased from 75% to 100% of WBCD. Although the setback for the other models can be significant, our model maintains nearly the same performance as the metrics values are slightly dropped (a drop of <0.5% for ACC, F1 score, and MCC, <1.5% for SPC, and <2.5% for PPV). Therefore, the results confirm the capability of our model in handling larger datasets without compromising its classification accuracy.

Finally, the rate of convergence of the competing optimizers is illustrated in Fig. 5. This figure demonstrates that GMVO has the best convergence performance among all optimizers. Specifically, GMVO reached the fitness value of 0.012, whereas CSA-ADAM and MVO got stuck in local minima, and hence converged to the fitness values of around 0.073 and 0.1 respectively. Although BAS-ADAM reached a fitness value of 0.0157, which

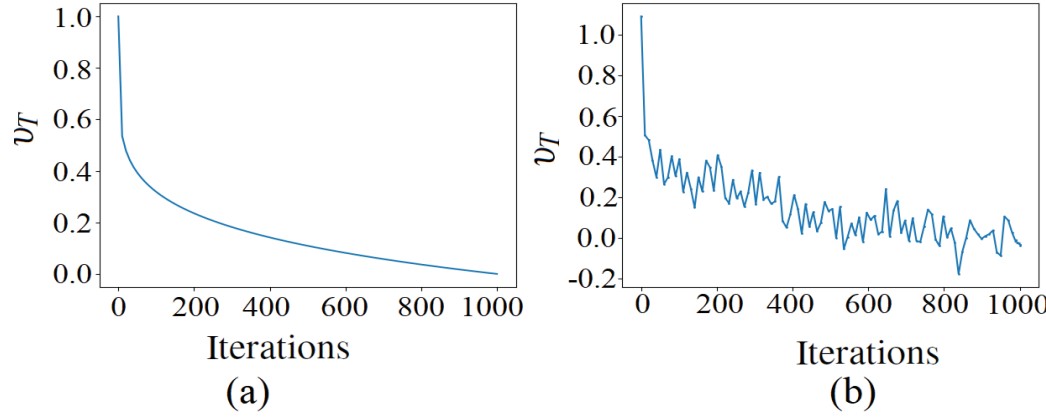

**Figure 6** The TDR over iterations of MVO and GMVO.

is quite close to the optimal point 0.012, but GMVO required less iterations to reach near that optimal point. Based on these findings, it can be deduced that GMVO can achieve a higher accuracy in a faster convergence rate compared to the other optimizers.

## DISCUSSIONS

In this section, we shed light on the key insights derived from the results presented in "Results" that obtained through a comprehensive analysis involving various performance metrics. These results demonstrate that MVO marginally improves the performance of DNN model compared to conventional optimizers, SGDM and ADAM. The reason for the insufficient improvement brought by such powerful metaheuristic is that, the large search space decreases the efficiency of MVO and, of course, similar metaheuristic algorithms. The results also reveal that the hybrid gradient-metaheuristic optimizers consistently outperforms both metaheuristic and gradient-based optimizers in training DNN models. This can be attributed to the improved global search capability of these hybrid optimizers that comes from integrating the benefits of gradient search rules (in moving towards the feasible regions and ignoring the unfeasible ones) with those of the population-based techniques (in escaping local optima).

From the results of the models based on hybrid optimizers, it can be inferred that the BAS-ADAM based model is more accurate than the one based on CSA-ADAM in WBCD, whereas both the models are competitive in WDBC. However, the standard deviation over the twenty runs is smaller with the CSA-ADAM based model in both WBCD and WDBC. These findings suggest that BAS-ADAM is able to converge to superior optima, but CSA-ADAM has more stable performance. On the other hand, the GMVO-based model is significantly more accurate than these two models in both WBCD and WDBC, making it the most appropriate one for clinical application. Moreover, GMVO outperforms BAS-ADAM and CSA-ADAM in terms of both time and iterations required to reach the optimal point. This makes GMVO is a promising optimizer for DNN learning, especially with high dimensional search space. The superiority brought by GMVO can be attributed to the benefits of MVO, such as good balance of exploration and exploitation, and the robust gradient search rule incorporated into MVO. The traveling distance rate over

iterations for original MVO and GMVO is depicted in Fig. 6. As can be seen, the traveling distance rate in GMVO is adjusted during the optimization based on the gradient of the objective function. This adaptation aims to allocate more resources to regions with significant changes in the objective function, facilitating effective exploration and exploitation of the solution space.

The main objective of this work is to improve the accuracy of DL models in BC detection and classification without an increase of complexity or computational burden. To this end, we devise a hybrid optimizer that enables an effective training for DL models and results in accurate prediction of BC with a small number of FNs and FPs. The implication of our findings is that, integrating the proposed GMVO-based model into real-world clinical workflows could augment the BC diagnostic capabilities of medical practitioners. The new model could enable early and accurate detection of BC, which can potentially improve patient outcomes and increase the survival rates.

## CONCLUSION

This article has proposed an efficient deep learning model for accurate detection and diagnosis of breast cancer. The new model has been effectively trained on WBCD and WDBC using a hybrid gradient-metaheuristic optimizer that incorporates a robust gradient search mechanism into the recently-proposed metaheuristic, multi-verse optimizer. The proposed hybrid optimizer integrates the benefits of gradient search rules (in moving towards the feasible regions and ignoring the unfeasible ones) with those of the metaheuristic techniques (in escaping local optima) to facilitate the global search capability in the high dimensional search space. The performance of the proposed optimizer in terms of training deep learning models for breast cancer detection and classification has been compared with that of original multi-verse optimizer, and the well-known optimizers, *i.e.*, SGDM and ADAM, as well as recently-proposed hybrid optimizers, namely, beetle ant search with ADAM, and cuckoo search algorithm with ADAM. The results reveal that, the original multi-verse optimizer does not bring significant improvement to the classification accuracy brought by the gradient-based optimizers, SGDM and ADAM, due to the huge search space. On the other hand, it has been observed that, compared to the other optimizers, the proposed optimizer improves the model's predictive accuracy in a statistically significant manner. Moreover, the results confirm the superiority of our optimizer over state-of-the-art optimizers in terms of both time and iterations required to reach the optimal point. These findings underscore the superior capabilities of our optimizer in training deep neural networks for breast cancer classification, making it a promising tool in optimization of machine learning models, especially with high dimensional search space. Despite the outstanding performance of our method, this study may be subject to some limitations. First, the application of the proposed model in a clinical environment and its versatility across diverse patient data, can impose new challenges and constraints to be considered. Second, there might be other metaheuristic

techniques that can be used to build hybrid optimizers with performance better than that of our optimizer. This fact follows the No Free Lunch theorem, which states that no single metaheuristic technique can perform superior in solving all optimization problems. This research, therefore, paves the way for future endeavors to explore and implement new hybrid gradient-metaheuristic optimizers in diverse applications, extending their impact across various domains of machine learning and data analysis.

Future studies can explore the potential of using our optimizer for training other neural network architectures, *e.g.*, CNN and RNN, on imaging modalities. Another possible direction for further research is to utilize the proposed method for the breast cancer stage classification in order to aid medical practitioners in making effective decisions for treatment.

### Funding
This work was supported by the National Natural Science Foundation of China (Grant Nos. 61672011 and 61472467). The funders had no role in study design, data collection and analysis, decision to publish, or preparation of the manuscript.

### Grant Disclosures
The following grant information was disclosed by the authors:
National Natural Science Foundation of China: 61672011 and 61472467.

### Competing Interests
The authors declare that they have no competing interests.

### Author Contributions
- Yassine EL kati conceived and designed the experiments, performed the experiments, analyzed the data, performed the computation work, prepared figures and/or tables, authored or reviewed drafts of the article, and approved the final draft.
- Shu-Lin Wang conceived and designed the experiments, performed the experiments, authored or reviewed drafts of the article, and approved the final draft.
- Mundher Mohammed Taresh analyzed the data, prepared figures and/or tables, and approved the final draft.
- Talal Ahmed Ali Ali conceived and designed the experiments, performed the experiments, performed the computation work, authored or reviewed drafts of the article, and approved the final draft.

### Data Availability
The Matlab code is available in the Supplemental File.
The Original Wisconsin Breast Cancer Database is available at https://doi.org/10.24432/C5HP4Z.

## Supplemental Information

Supplemental information for this article can be found online at http://dx.doi.org/10.7717/peerj-cs.2578#supplemental-information.

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
