# Peer review of "Enhancing breast cancer diagnosis using deep learning and gradient multi-verse optimizer: a robust biomedical data analysis approach"

_PeerJ Computer Science, doi:10.7717/peerj-cs.2578_

## Round 0.1 · original submission · Major Revisions

Dear authors,

The reviews for your manuscript are included at the bottom of this letter. According to this reviews, your article has not been recommended for publication in its current form. However, we do encourage you to address the concerns and criticisms of the reviewers and resubmit your article once you have updated it accordingly.

When submitting the revised version of your article, it will be better to address the following:

1. The abstract does not present the creation or usage of the dataset.
2. The problem statement is not clear in the Introduction. This section should include a problem definition, motivation, overview of the proposed solution, enumeration of the contributions and organization of the paper. Organization of the paper should be given at the end of Introduction section.
3. Evaluation of the techniques covered in the literature is missing. Negative and positive aspects of these techniques should be stated. The lessons learned from the literature is absent. These are the key aspects to motivate for the research and for new researchers who want to tackle the same problem.
4. All reported graphs should be accompanied by some concrete description of the lessons learned from the results reflected in the graph. It is important to explain them in detail and to enrich them with some semantics by showing the reasons for these results, how they can be further improved, etc.
5. In the conclusions, please state explicitly what lessons can be learnt from this study and then describe in more detail the future research directions.
6. English grammar and writing style errors should be corrected. There are many writing errors that should be corrected.

Best wishes,

·

Basic reporting

1- Clarity and Ambiguity:
• Inconsistency in Terminology: The article employs the terms "GMOV" and "GMVO" interchangeably. Consistency is essential for achieving clarity.
 Improvement: Employ a solitary and uniform term throughout the entirety of the article. For example, if "GMVO" is the accurate terminology, make sure it is employed consistently.
• Verbose Sentences: Certain sentences can be lengthy and intricate, rendering them difficult to comprehend.
• Improvement: Divide intricate sentences into shorter, more easily comprehensible ones.
2- Professional English:
• Informal Phrasing: Some expressions are relatively casual for a technical document.
• Improvement: Use more formal language.
3- Insufficient Background and Context:
• Lack of Background Information: The article lacks sufficient context regarding the present condition of optimization algorithms for deep neural networks and their utilization in breast cancer classification.
• Improvement: Provide an elaborate introduction that summarizes the current research in the field, identifies the obstacles encountered, and highlights the importance of enhancing optimization algorithms for deep neural networks in medical diagnosis.
4- Literature References:
• Missing References: The current research lacks substantial references to prior work that establish its foundation.
• Improvement: Provide citations for studies that have played a significant role in the advancement of optimization algorithms and their utilization in deep neural networks (DNNs).
5- Figures and Tables:
• Insufficient Description and Labeling: The text lacks sufficient description or labeling of the figures and tables mentioned.
• Improvement: Every figure and table must be cited in the text along with a concise explanation of its content and importance.
6- Explicit Hypothesis Statement:
• Lack of Clear Hypothesis: The article lacks a clear statement of the hypothesis or research question that the study intends to investigate.
• Improvement: Clearly express the hypothesis or research question in a concise manner within the introduction.
7- Absence of Proofs: The article lacks comprehensive demonstrations for the theorems or assertions presented.
• Improvement: Present comprehensive evidence to support all theorems or significant assertions. If the study asserts that the GMVO optimizer enhances accuracy to a significant degree, it is imperative to provide mathematical or statistical evidence to substantiate this assertion.

Experimental design

1- Research Question Definition:
• Lack of Explicit Research Question: The article lacks a clearly stated research question at the beginning.
• Improvement: Commence the article by formulating a research question that is unambiguous and well-defined. For instance, what is the effect of incorporating a gradient-based search mechanism into the Multi-Verse Optimizer (MVO) on the efficacy of training deep neural networks for breast cancer classification?
2- Identifying Knowledge Gap:
• Unclear Identification of Knowledge Gap: The article does not clearly specify a particular deficiency in the current body of research that the study intends to address.
• Improvement: Precisely express the deficiency in existing research. For instance, even though there have been improvements in deep learning optimization methods, there is still a requirement for efficient algorithms that can enhance classification accuracy without imposing a substantial computational burden. This study aims to fill this gap by suggesting a new incorporation of gradient-based techniques into the MVO framework.
3- Technical Detail Omissions:
• Missing Technical Aspects: The user's text lacks detailed explanations of crucial technical elements, including any preprocessing techniques applied to the WBCD dataset, the specific architecture of the deep neural network used, and the incorporation of gradient-based adjustments into the MVO framework.
 Improvement: Provide a clear and detailed explanation of each preprocessing step, including the architecture details such as the number of layers and activation functions used. Additionally, explicitly describe how gradients were computed and utilized within the MVO framework.

Validity of the findings

1- Lack of Impact Assessment:
• Impact Not Clearly Stated: The article does not explicitly evaluate the influence or originality of the proposed optimization algorithm in comparison to existing methods.
 Improvement: Explicitly state in the introduction or discussion sections the reasons why the proposed algorithm is a substantial improvement compared to existing methods. Emphasize particular elements such as performance metrics, computational efficacy, or suitability for wider datasets or domains.
2- Novelty and Advance:
• Novelty Not Clearly Demonstrated: Although the article discusses enhancements in precision and the rate at which the model reaches a solution, it fails to adequately emphasize the unique aspects or the substantial progress of the proposed methodology within the field.
 Improvement: Explain the uniqueness of incorporating gradient-based search into the MVO framework in a comprehensive manner. Emphasize distinctive characteristics, novel approaches, or modifications that set it apart from previous methodologies.
3- Clarity and Connection to Research Question:
• Lack of Direct Linkage: The conclusions should explicitly relate to the initial research question and clearly articulate how they address or contribute to addressing that question.
 Improvement: Make sure that every conclusion directly relates to the research question that was initially presented at the start of the study. To illustrate, explain how the observed enhancements in precision and rate of convergence directly target the primary objective of optimizing the training of deep neural networks for the classification of breast cancer.

Additional comments

General comments:-

- Conclusion and Future Directions: The conclusion succinctly encapsulates the findings and underscores the possible ramifications of the research. To enhance this section, contemplate elaborating on prospective avenues for investigation, encompassing potential implementations in alternative domains or expansions to diverse categories of neural networks or datasets.
Overall, the article demonstrates potential in its methodology and discoveries. By addressing these comments, the research can be improved in terms of clarity, rigor, and impact, making it more persuasive for publication in the desired journal.

Reviewer 2 ·

Basic reporting

1. The paper contains several spelling mistakes. For example it's written 'Methods and Matrials' as a section-title.
2. The paper should be thoroughly checked for grammatical errors.
3. The paper presents the background on the deep-learning based appoaches in general, but lacks the overview of how deep-learning has been applied to medical imaging and diagnosis, specifically in the field of breast cancer diagnosis.
4. The introduction part should clearly display the motivation of using the proposed approach for the medical dataset, alongwith explicitly stating the challenges and limitations of current breast cancer diagnostic methods.

Experimental design

1. The experimental design of the paper is satisfactory but requires several enhancements such as description of all hyperparameters, detailed statistical analysis, etc.
2. The paper lacks proper theoretical justification for why the proposed approach should outperform current benchmarks.
3. The study uses only the WBCD dataset. The experiments should be performed with other medium to large datasets to validate the generalizability of the results.
4. Figure 1 can be explained more clearly.

Validity of the findings

1. The results show performance improvements for the WBCD dataset. However, the novelty of the approach is not thoroughly assessed. There is a lack of detailed comparative analysis with existing state-of-the-art methods for example in terms of scalability,etc.
2. The conclusions are drawn from the single dataset, which may not represent the full complexity of the problem. Additional experiments should be performed and the findings should be reported with respect to the new datasets.
3. The paper does not clearly mention the actual clinical implications of the findings.

Additional comments

-

Reviewer 3 ·

Basic reporting

See below

Experimental design

See below

Validity of the findings

See below

Additional comments

-The abstract part needs to explain the writing motivation and research process of the article more clearly
-The parameter definition of the formula needs to be clearer
-The conclusion part needs to explain the research limitations of the paper.
-The author should highlight his contribution in this article.
-The author should describe the differences between the proposed method and other methods.
- The inspiration of your work must be highlighted via appropriate citations
-How is the optimality of the proposed method against compared methods? How is better the quality of your method compared to others?

---

## Round 0.2 · accepted · Accept

Dear Authors,

Thank you for the revised paper. The reviewers think that your paper can accepted in this current form. The manuscript seems ready for publication.

Best wishes,

·

Basic reporting

My comments have been addressed. It is acceptable in the present form.

Experimental design

My comments have been addressed. It is acceptable in the present form.

Validity of the findings

My comments have been addressed. It is acceptable in the present form.

Additional comments

The article meets the requirements of the journal and can be published in its current state.

Reviewer 3 ·

Basic reporting

ok

Experimental design

ok

Validity of the findings

ok